# Antitumor Activity of an Anti-EGFR/HER2 Bispecific Antibody in a Mouse Xenograft Model of Canine Osteosarcoma

**DOI:** 10.3390/pharmaceutics14112494

**Published:** 2022-11-17

**Authors:** Nami Tateyama, Hiroyuki Suzuki, Tomokazu Ohishi, Teizo Asano, Tomohiro Tanaka, Takuya Mizuno, Takeo Yoshikawa, Manabu Kawada, Mika K. Kaneko, Yukinari Kato

**Affiliations:** 1Department of Antibody Drug Development, Tohoku University Graduate School of Medicine, 2-1 Seiryo-machi, Aoba-ku, Sendai 980-8575, Miyagi, Japan; 2Department of Molecular Pharmacology, Tohoku University Graduate School of Medicine, 2-1 Seiryo-machi, Aoba-ku, Sendai 980-8575, Miyagi, Japan; 3Institute of Microbial Chemistry (BIKAKEN), Numazu, Microbial Chemistry Research Foundation, 18-24 Miyamoto, Numazu-shi 410-0301, Shizuoka, Japan; 4Institute of Microbial Chemistry (BIKAKEN), Laboratory of Oncology, Microbial Chemistry Research Foundation, 3-14-23 Kamiosaki, Shinagawa-ku, Tokyo 141-0021, Japan; 5Laboratory of Molecular Diagnostics and Therapeutics, Joint Faculty of Veterinary Medicine, Yamaguchi University, 1677-1, Yoshida 753-8515, Yamaguchi, Japan; 6Department of Pharmacology, Tohoku University Graduate School of Medicine, 2-1 Seiryo-machi, Aoba-ku, Sendai 980-8575, Miyagi, Japan

**Keywords:** EGFR, HER2, bispecific antibody, ADCC, CDC, canine osteosarcoma

## Abstract

The overexpression of epidermal growth factor receptors (EGFRs) has been reported in various human tumors, including breast, gastric, lung, colorectal, and pancreatic cancers. Humanized anti-EGFR and anti-human epidermal growth factor receptor 2 (HER2) monoclonal antibodies (mAbs) have been shown to improve patients’ survival. Canine tumors resemble human tumors in the initiation and progression. We previously established a defucosylated mouse-dog chimeric anti-EGFR mAb (E134Bf) and a mouse-dog chimeric anti-HER2 mAb (H77Bf), which exerted antitumor activities in canine tumor xenograft models. Here, we produced E134Bf antibody fused to H77Bf single chain Fv at the light chains (E134Bf-H77scFv). The bispecific E134Bf-H77scFv recognized dog EGFR (dEGFR) and dog HER2 (dHER2)-overexpressed Chinese hamster ovary-K1 cells by flow cytometry. E134Bf-H77scFv also reacted with dEGFR/dHER2-positive canine osteosarcoma D-17 cells, and possesses a high binding-affinity (*K*_D_: 1.3 × 10^−9^ M). Furthermore, E134Bf-H77scFv exerted antibody-dependent cellular cytotoxicity and complement-dependent cytotoxicity against D-17 cells in the presence of canine mononuclear cells and complement, respectively. Moreover, administration of E134Bf-H77scFv suppressed the development of D-17 xenograft tumor in mice early compared with the control dog IgG, E134Bf and H77Bf alone. These results indicate that E134Bf-H77scFv exerts antitumor activities against dEGFR/dHER2-positive canine tumors, and could be a valuable treatment regimen for canine tumors.

## 1. Introduction

Therapeutic monoclonal antibodies (mAbs) for tumors have revolutionized over several decades due to the development of the hybridoma technique [1] and the mAb humanization technique [2]. Currently, more than 100 mAbs have been approved by the US Food and Drug Administration (FDA) for the treatment of not only tumors, but also inflammatory diseases [3]. MAbs can bind to target antigens with high specificity, and inhibit tumor promoting growth factor signaling pathways through their neutralization effects. In addition, mAbs potentiate the host immune responses including antibody dependent cellular cytotoxicity (ADCC), complement dependent cytotoxicity (CDC), and antibody-dependent cellular phagocytosis (ADCP) [4]. Due to the successful application of immunoglobulin G (IgG) type mAbs, several antibody formats including bispecific antibodies and antibody–drug conjugates (ADCs) have been approved by the FDA as next generation therapeutic agents for various types of tumors [5,6].

Targeting receptor tyrosine kinase signaling pathways is one of the important strategies for tumor therapy [7]. Human epidermal growth factor receptor (HER) families belong to type I transmembrane glycoproteins that are overexpressed in various solid tumors [8]. Through their homo- and hetero-dimerization, a broad repertoire of oncogenic signaling pathways is activated, and these are essential for tumor development [9]. Anti-epidermal growth factor receptor (EGFR) mAb, cetuximab, can block EGFR activation through the binding to its ligand binding domain and, thus, inhibit the EGFR signaling pathways. Moreover, cetuximab (IgG_1_ isotype) exhibits ADCC activity, and directs cytotoxic immune effector cells to EGFR-positive tumor cells [10,11]. Cetuximab was approved in combination with platinum-based chemotherapy, showing good outcomes including overall survival, response rate, and progression-free survival in metastatic/recurrent head and neck squamous cell carcinomas [12]. Trastuzumab, the first FDA-approved anti-HER2 mAb, has been the most effective therapy for HER2-positive breast cancers and later in HER2-positive gastric cancers for more than 20 years [13]. Clinically, the efficacy is mediated by the involvement of immunologic engagements, including ADCC, CDC, and ADCP activity [4].

With the increase in lifespan of both humans and dogs, the tumor incidence has increased, as well. Spontaneous canine tumors closely look like human tumors [14]. Osteosarcoma is the most common primary bone tumor in dogs. The rate of osteosarcoma in dogs is 27 times higher than that in humans, and lung metastasis is frequently observed. In fact, a one-year survival rate is only 45% [15]. Currently, treatment options for osteosarcoma in dogs include surgery, radiation, and chemotherapy [16,17]. However, sufficient therapeutic efficacies have not been reported. Therefore, the development of other therapeutic options for canine osteosarcoma is necessary. Specifically, EGFR expression was significantly upregulated in osteosarcoma lung metastasis compared to the extrapulmonary sites [18]. Doyle et al. reported the vaccine therapy using dog EGFR (dEGFR)/dog HER2 (dHER2) peptide in canine osteosarcoma. Dogs with osteosarcoma were immunized with a dEGFR/dHER2 peptide vaccine, which resulted in an increased anti-dEGFR/dHER2 antibody production and T cells infiltration into tumors. The vaccinated dogs with osteosarcoma showed tumor regression and increased survival. These data support the approach of amplifying antitumor immunity by dEGFR/dHER2 peptide [19].

The ADCC is activated by the binding of the FcγRIIIa on natural killer (NK) cells to the Fc region of mAbs [4]. However, the *N*-linked glycosylation in the Fc region is reported to reduce the binding to the FcγRIIIa on NK cells [20,21]. Fucosyltransferase 8 (FUT8) is an important enzyme that mediates the *N*-linked glycosylation (core fucosylation) in the Fc region [21]. Therefore, FUT8-knockout Chinese hamster ovary (CHO) cells have been shown to produce completely defucosylated recombinant antibodies. The defucosylated mAb strongly binds to FcγRIIIa and exerts potent ADCC activity when compared to a conventional mAb [22].

Previously, we developed an anti-EGFR mAb (EMab-134) [23] and anti-HER2 mAb (H_2_Mab-77) [24] using the Cell-Based Immunization and Screening method. Subsequently, we produced mouse-dog chimeric anti-EGFR mAb (E134B) and anti-HER2 mAb (H77B) from the information of variable regions of EMab-134 and H_2_Mab-77, respectively. Furthermore, we evaluated the antitumor effect of their respective defucosylated forms, E134Bf [25,26] and H77Bf [27] against canine tumors in mouse xenograft models. In this study, we produced a defucosylated bispecific Ab against dEGFR and dHER2, and evaluated the ADCC, CDC, and antitumor activities in canine osteosarcoma D-17 xenograft.

## 2. Materials and Methods

### 2.1. Cell Lines

CHO-K1 and a canine osteosarcoma cell line D-17 were purchased from the American Type Culture Collection (Manassas, VA, USA). Stable dog EGFR-overexpressed CHO-K1 (CHO/dEGFR) [26] and dog HER2-overexpressed CHO-K1 (CHO/dHER2) [27] were established as described previously. CHO-K1, CHO/dEGFR, and CHO/dHER2 were cultured in RPMI-1640 medium (Nacalai Tesque, Inc., Kyoto, Japan), containing 10% fetal bovine serum (FBS; Thermo Fisher Scientific, Inc., Waltham, MA, USA), 100 μg/mL streptomycin, 100 units/mL of penicillin, and 0.25 μg/mL amphotericin B (Nacalai Tesque, Inc.). D-17 was cultured in MEM medium (Nacalai Tesque, Inc.), supplemented with 10% FBS, 1 mM of sodium pyruvate, 100 units/mL of penicillin, 100 μg/mL streptomycin, and 0.25 μg/mL amphotericin B.

### 2.2. Animals

In all animal experiments, we followed the guidelines of the Institutional Committee for Experiments of the Institute of Microbial Chemistry (Numazu, Japan) (approval no. 2022-036). Mice were bred in a specific pathogen-free condition on an 11 h light/13 h dark cycle. Mice weight were monitored two times per week and health were monitored three times per week. The loss of original body weight was determined to a point >25% and/or a maximum tumor size >2000 mm^3^ and/or significant changes in the appearance of tumors as humane endpoints for euthanasia. Cervical dislocation was used for euthanasia. Mice death was confirmed by respiratory arrest and rigor mortis.

### 2.3. Antibodies

Previously, we have established mouse-dog chimeric anti-EGFR mAb (E134B) [26] and anti-HER2 mAb (H77B) [27]. In this study, to generate E134Bf-H77scFv, we first constructed a single chain Fv of H77B (H77scFv) by connecting the V_H_ and V_L_ cDNA of H77B with a linker sequence (GGGGSGGGGSGGGGS). The H77scFv cDNA was further fused at the 3′ end of the light chain cDNA of E134B. The cDNA of the E134B heavy chain and E134B light chain-H77scFv were transduced into BINDS-09 (FUT8 knockout ExpiCHO-S) cells using the ExpiCHO Expression System (Thermo Fisher Scientific, Inc.) to produce the defucosylated form [25,26,27,28,29,30,31,32]. A defucosylated and bispecific antibody, E134Bf-H77scFv, was purified using Ab-Capcher (ProteNova Co., Ltd., Kagawa, Japan). We confirmed their purity by SDS-PAGE (Appendix A). Dog IgG was purchased from Jackson ImmunoResearch Laboratories, Inc. (West Grove, PA, USA).

### 2.4. Flow Cytometry

CHO-K1, CHO/dEGFR, CHO/dHER2, and D-17 cells were suspended in blocking buffer [0.1% bovine serum albumin in phosphate-buffered saline (PBS)] containing 1 μg/mL of E134Bf, H77Bf, E134Bf-H77scFv for 30 min at 4 °C. Then, cells were suspended in blocking buffer containing Alexa Fluor 488-conjugated anti-dog IgG (1:1000; Jackson ImmunoResearch Laboratories, Inc.) for 30 min at 4 °C. Fluorescence data were collected by the Cell Analyzer EC800, and analyzed by EC800 software ver. 1.3.6 (Sony Corp., Tokyo, Japan).

### 2.5. Determination of Binding Affinity

Serially diluted E134Bf (0.0006–10 μg/mL), H77Bf (0.0006–10 μg/mL), and E134Bf-H77scFv (0.0006–10 μg/mL) were suspended with CHO/dEGFR, CHO/dHER2, and D-17 cells. The cells were further treated with Alexa Fluor 488-conjugated anti-dog IgG (1:100). Fluorescence data were obtained by the Cell Analyzer EC800. To determine the apparent dissociation constant (*K*_D_), GraphPad Prism version 8 (the fitting binding isotherms to built-in one-site binding models, GraphPad Software, Inc., La Jolla, CA, USA) was used.

### 2.6. ADCC

Induction of ADCC by E134Bf, H77Bf, and E134Bf-H77scFv was assayed as follows. Canine mononuclear cells (MNCs, from Yamaguchi University) were resuspended in DMEM supplemented with 10% FBS, and were used as effector cells. Target D-17 cells were labeled with 10 μg/mL of Calcein AM (Thermo Fisher Scientific, Inc.) [25,26,27,28,29,30,31,32,33,34,35,36,37,38,39,40,41,42]. The effector canine MNCs and the target cells (effector/target cells ratio, 50) were incubated in the presence of 100 μg/mL of control dog IgG, E134Bf, H77Bf, and E134Bf-H77scFv for 4.5 h at 37 °C. The Calcein release into the medium was measured with an excitation wavelength (485 nm) and an emission wavelength (538 nm) using a microplate reader (Power Scan HT; BioTek Instruments, Inc., Winooski, VT, USA). Cytolyticity (% lysis) was determined as described previously [27].

### 2.7. CDC

The target D-17 cells were incubated with 10 μg/mL calcein AM [25,26,27,28,29,30,31,32,33,34,35,36,37,38,39,40,41,42], and were incubated with rabbit complement (Cedarlane Laboratories, Hornby, Ontario, Canada; final dilution 1:10) in the presence of 100 μg/mL of control dog IgG, E134Bf, H77Bf, and E134Bf-H77scFv. The Calcein release into the medium was determined as indicated in the previous section. Cytolyticity (% lysis) was determined as described previously [27].

### 2.8. Antitumor Activities in Xenografts of D-17

BALB/c nude mice (female, 4 weeks old) were purchased from Charles River Laboratories, Inc. D-17 cells (5 × 10^6^ cells) were subcutaneously inoculated into the left flank of mice together with BD Matrigel Matrix Growth Factor Reduced (BD Biosciences, San Jose, CA, USA). On day 8 after the inoculation, 100 μg of E134Bf, H77Bf, and E134Bf-H77scFv, and control dog IgG (*n* = 8) in PBS (100 μL) were intraperitoneally injected. Additional antibodies were intraperitoneally injected on days 15 and 22. Furthermore, on days 8, 15, and 22, canine MNCs (5 × 10^5^ cells) were injected into the surrounding tumors. The tumor volume was measured on days 8, 10, 15, 17, 22, and 25, as described previously [25,26,27,28,29,30,31,32,33,34,35,36,37,38,39,40,41,42,43].

## 3. Results

### 3.1. Flow Cytometric Analysis against dEGFR and dHER2-Expressing Cells Using E134Bf-H77scFv

We previously developed mouse-dog chimeric mAbs including E134B (anti-dEGFR) and H77B (anti-dHER2), both of which possess the B type dog IgG backbone to confer the ADCC activity. We further produced the corresponding defucosylated forms, E134Bf [26] and H77Bf [27], respectively. To generate a bispecific Ab against dEGFR and dHER2, we first constructed the cDNA of H77scFv, following the ligation to the light chain cDNA of E134B (Figure 1). Subsequently, we transduced the cDNAs of E134B heavy chain and the E134B light chain-H77scFv into FUT8 knockout ExpiCHO-S cells, and purified the defucosylated bispecific Ab against dEGFR and dHER2, which is labeled as E134Bf-H77scFv.

We first confirmed the reactivity of E134Bf-H77scFv to dEGFR and dHER2 expressed cells using flow cytometry. As shown in Figure 2, E134Bf-H77scFv exhibited the similar reactivities to CHO/dEGFR (Figure 2A) and CHO/dHER2 (Figure 2B) cells compared with E134Bf and H77Bf, respectively. In contrast, E134Bf-H77scFv never reacted with parental CHO-K1 cells (Figure 2C). Thereafter, we examined the reactivity of E134Bf-H77scFv against dEGFR and dHER2-positive canine osteosarcoma D-17 cells. As shown in Figure 2D, E134Bf-H77scFv could recognize D-17 cells. Then, we determined apparent dissociation constants *(K*_D_) of E134Bf, H77Bf and E134Bf-H77scFv to D-17 cells using flow cytometry. As shown in Figure 3, the *K*_D_ values for the interaction of E134Bf, H77Bf, and E134Bf-H77scFv with D-17 cells were 6.0 × 10^−^^10^ M, 2.9 × 10^−^^10^ M, and 1.3 × 10^−^^9^ M, respectively.

### 3.2. E134Bf-H77scFv-Mediated ADCC and CDC in D-17 Cells

We next investigated whether E134Bf-H77scFv was capable of mediating ADCC against D-17 cells. E134Bf-H77scFv showed ADCC (15.2% cytotoxicity) against D-17 cells more effectively than the control dog IgG (4.1% cytotoxicity; *p* < 0.01). There was no significant difference between E134Bf or H77Bf and the control dog IgG against D-17 in this experimental condition (Figure 4A). We next examined whether E134Bf-H77scFv could exert CDC against D-17 cells. As shown in Figure 4B, H77Bf, E134Bf, and E134Bf-H77scFv induced a higher degree of CDC (33.6% [*p* < 0.05], 40.7% [*p* < 0.01], and 44.4% [*p* < 0.01] cytotoxicity, respectively) in D-17 cells compared with that induced by the control dog IgG (13.8% cytotoxicity). These results indicated that E134Bf-H77scFv exhibited higher levels of ADCC and CDC activities against D-17 cells.

### 3.3. Antitumor Effects of E134Bf-H77scFv in the Mouse Xenograft of D-17 Cells

To evaluate the antitumor activity against D-17 xenograft tumor, E134Bf-H77scFv, E134Bf, H77Bf and control dog IgG were intraperitoneally injected into mice on days 8, 15, and 22 after the inoculation of D-17 cells. Furthermore, on days 8, 15, and 22, canine MNCs were injected surrounding the tumors. On days 8, 10, 15, 17, 22, and 25 after the inoculation, the tumor volume was determined. The E134Bf-H77scFv administration resulted in faster reduction of tumor volume on days 10 (*p* < 0.05), 15 (*p* < 0.05), and 17 (*p* < 0.01) than that of the E134Bf, H77Bf and control dog IgG (Figure 5A). However, on days 22 and 25, significant reduction of tumor was also observed in E134Bf and H77Bf treated groups as well as the E134Bf-H77scFv treated group compared with the controls (Figure 5A). As shown in Figure 5B, the weight of D-17 tumors treated with E134Bf-H77scFv, E134Bf, and H77Bf was significantly lower than that treated with control dog IgG (70%, 64%, 58% reduction, respectively; *p* < 0.01). D-17 tumors that were resected from mice on day 25 are shown in Figure 5C. The loss of body weight was not observed in each group (Figure 5D). The mice on day 25 are shown in Figure 5E.

## 4. Discussion

In this study, we developed a novel bispecific antibody E134Bf-H77scFv against dEGFR and dHER2 (Figure 1), and showed more potent ADCC activity compared with that of E134Bf and H77Bf (Figure 4A). Furthermore, E134Bf-H77scFv showed the antitumor activity against D-17 xenografts at earlier periods compared with E134Bf and H77Bf treatments (Figure 5A), suggesting that E134Bf-H77scFv possesses the different mode of actions. As shown in Figure 2, E134Bf-H77scFv recognized dEGFR and dHER2-overexpressed CHO-K1 cells. Furthermore, E134Bf-H77scFv also reacted with dEGFR and dHER2 double positive D-17 cells. However, we have not determined whether E134Bf-H77scFv can induce the cross-link between dEGFR and dHER2 on the cell surface. In our epitope mapping analysis, EMab-134, the original mAb of E134B, recognized the sequence (_377_-RGDSFTHTPP-_386_) in domain III of EGFR [44]. The EGFR peptide (amino acids 375-394) containing the epitope potently inhibited the recognition of D-17 cells by E134Bf. In the same condition, the inhibition to E134Bf-H77scFv was weaker than that to E134Bf (Appendix A), suggesting that E134Bf-H77scFv reacts with dHER2 on D-17 cells. However, the epitope of H77scFv has not been identified. Further investigations are essential to reveal the structural relevance of recognition, and the mechanism of the antitumor activity.

E134Bf-H77scFv is a novel modality for targeting dEGFR and dHER2. Because H77scFv are fused to the light chain of E134B but not to the heavy chain, E134Bf-H77scFv recognizes dual antigens without affecting the Fc-mediated ADCC activity. Furthermore, in case of bispecific antibody with two unique Fab domains, the combination of V_H_ and V_L_ cDNAs should be considered, and the selection of objective bispecific antibody is required. In contrast, E134Bf-H77scFv can be produced by transfection with single combination of V_H_ and V_L_ cDNA, and purified by the conventional Fc binding beads.

Previously, different types of dual-targeting EGFR and HER2 bispecific modalities have been tested in preclinical studies. A bispecific affibody molecule to EGFR or HER2 was generated by linking a bivalent HER2-binding affibody to a bivalent EGFR-binding affibody with a linker sequence (Gly4-Ser)3. The bispecific affibody was shown to bind to both HER2-overexpressed SKBR-3 and EGFR-overexpressed A-431 cells [45]. “AffiMabs” were generated by fusing an EGFR-targeting affibody molecule to trastuzumab’s heavy or light chains. AffiMabs were shown to induce the downregulation of both EGFR and HER2. Furthermore, AffiMabs exhibited a more potent anti-proliferative effect than trastuzumab in vitro [46]. Moreover, a bispecific nanobody targeting EGFR and HER2 (bsNb) was developed. The bsNb was further ligated to rhamnose (Rha) hapten to reconstitute the Fc effector biological function. The bsNb-Rha retained dual-targeting activity to EGFR and HER2, and elicited potent antitumor effects in vivo via the Fc-mediated engagement of endogenous anti-Rha antibodies [47]. However, bispecific antibody formats against EGFR and HER2 have not been applied in the clinic at present. Studies have demonstrated that overexpression of EGFR family proteins is observed in aggressive canine tumors including osteosarcoma [48] and hemangiosarcoma [49] with shortened the survival. We have already showed that EMab-134 and H_2_Mab-77, original antibodies of E134Bf-H77scFv, can be used in immunohistochemistry [23,24]. Therefore, EMab-134, H_2_Mab-77, and E134Bf-H77scFv are expected to be used in clinics for both the diagnosis and treatment of canine tumors.

Bispecific antibodies have been approved as therapeutic agents to cover unmet clinical needs [50,51]. A wide range of bispecific antibody formats has been developed for cancer therapy through different mechanisms including the engagement of T cells or other immune cells (e.g., NK cells) to induce antitumor immunity, and bridging receptors to inhibit or activate the downstream signaling pathways [52]. For targeting receptor kinase signaling pathways including EGFR, HER2, HER3, and MET in tumor cells, GBR1302 has been developed as a T-cell engager to direct CD3-positive T cells to HER2 on tumor cells [53]. Furthermore, bispecific antibodies that recognize different receptors involved in signaling crosstalk can be used to avoid bypass signal transduction during tumor development [54,55,56]. Growing number of evidence has indicated that MET signaling contributes to the resistance of EGFR tyrosine kinase inhibitors in non-small cell lung cancers (NSCLC) [57,58,59]. Therefore, dual inhibition of aberrant oncogenic signaling has become a promising strategy. Amivantamab is a dual-targeting EGFR and MET bispecific antibody that can inhibit the EGFR and MET signaling pathways [56]. In various tumor models, amivantamab efficiently downregulates the expression of EGFR and MET, and exhibited antitumor immunity with Fc-mediated effector interactions [60]. The FDA granted accelerated approval to JNJ-61186372 (Amivantamab) for the treatment for locally advanced or metastatic NSCLC. Another example is MCLA-128, a bispecific antibody for HER2 and HER3. MCLA-128 can inhibit heregulin (a HER3 ligand)-mediated signaling of HER2/HER3, and suppress tumor cell survival and proliferation via the downregulation of PI3K/Akt signaling [61]. Clinical studies on MCLA-128 are ongoing in patients with breast cancer, pancreatic cancer, and NSCLC.

For the development of E134Bf-H77scFv to treat canine tumors in the clinic, the effect of E134Bf-H77scFv on dEGFR homodimer and dEGFR/dHER2 heterodimer-mediated signaling should be investigated. Further, it should be determined whether E134Bf-H77scFv can induce the internalization of dEGFR and dHER2. Canine tumors represent an outbred population, and resemble human tumors in the initiation, disease progression, growth factors and oncogene mutations. A growing number of evidences has suggested that elevated expression of other HER family and their ligands confer the resistance to trastuzumab [62]. For instance, co-expression of EGF in HER2-positive cell lines reduced the growth inhibitory effect of trastuzumab by modulating EGFR/HER2 interaction [63]. Furthermore, EGFR expression was associated with worse prognosis, specifically when co-expressed with HER2 in advanced breast cancer [64]. Therefore, future investigations could contribute to not only canine, but also human EGFR/HER2-positive tumor treatment.

## Figures and Tables

**Figure 1 pharmaceutics-14-02494-f001:**
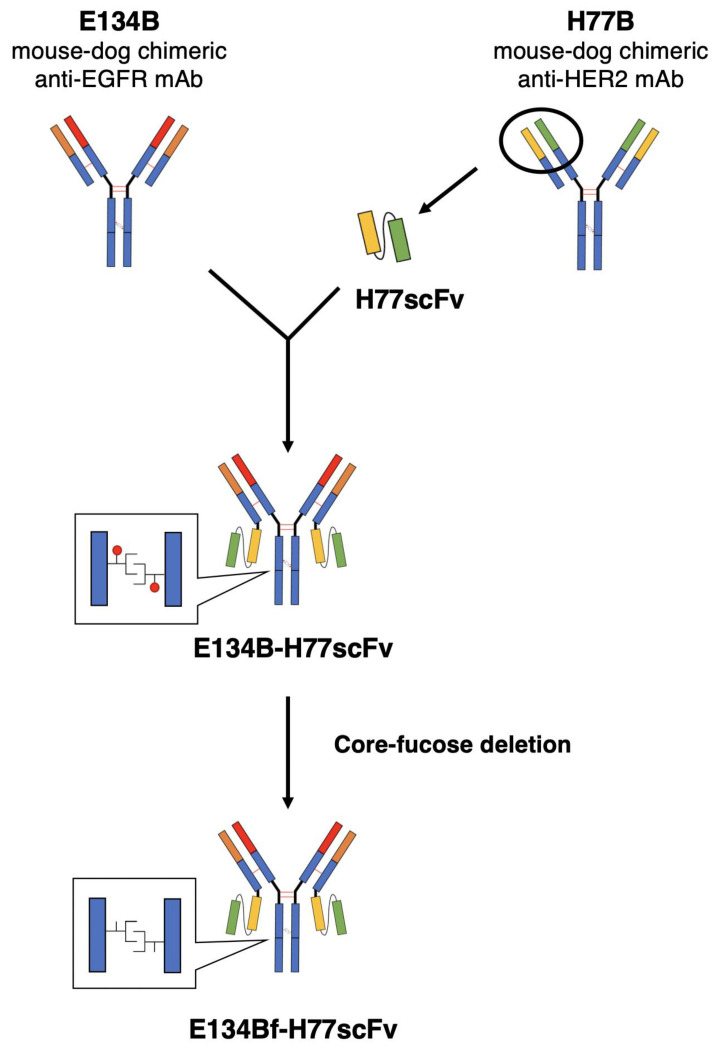
Schematic illustration of the production of bispecific antibody, E134Bf-H77scFv from E134B and H77B.

**Figure 2 pharmaceutics-14-02494-f002:**
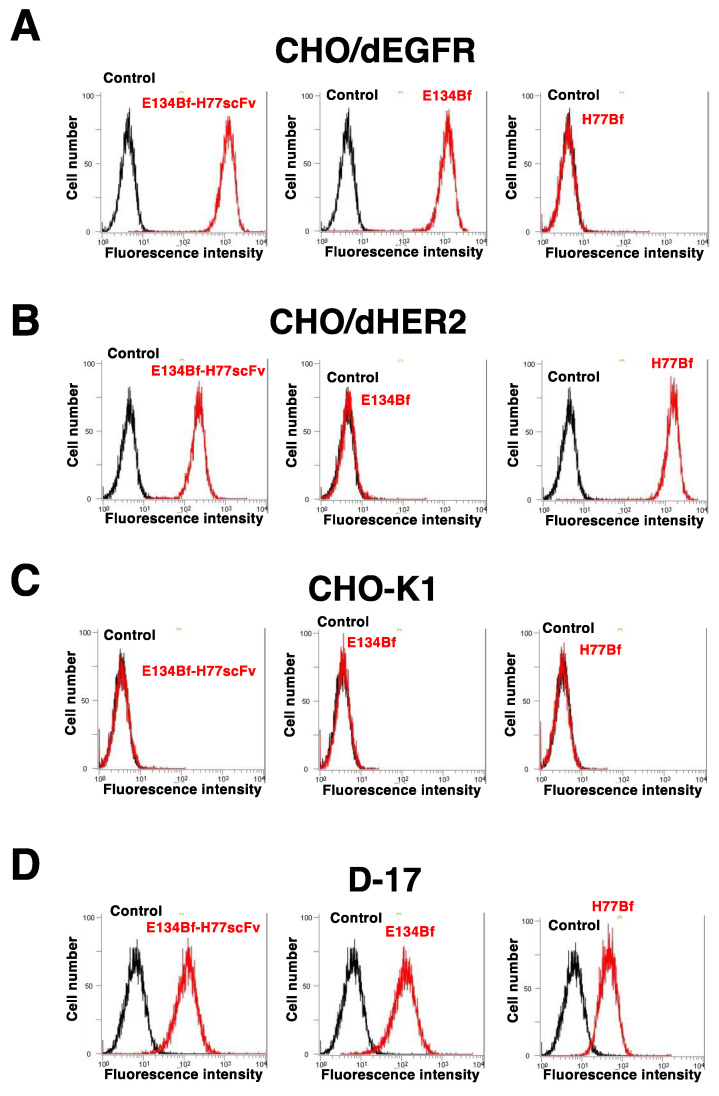
Flow cytometry using E134Bf, H77Bf, and E134Bf-H77scFv. CHO/dEGFR (**A**), CHO/dHER2 (**B**), CHO-K1 (**C**), and D-17 (**D**) cells were treated with 1 μg/mL of E134Bf, H77Bf, and E134Bf-H77scFv or buffer control, followed by Alexa Fluor 488-conjugated anti-dog IgG. Fluorescence data were analyzed using the Cell Analyzer EC800.

**Figure 3 pharmaceutics-14-02494-f003:**
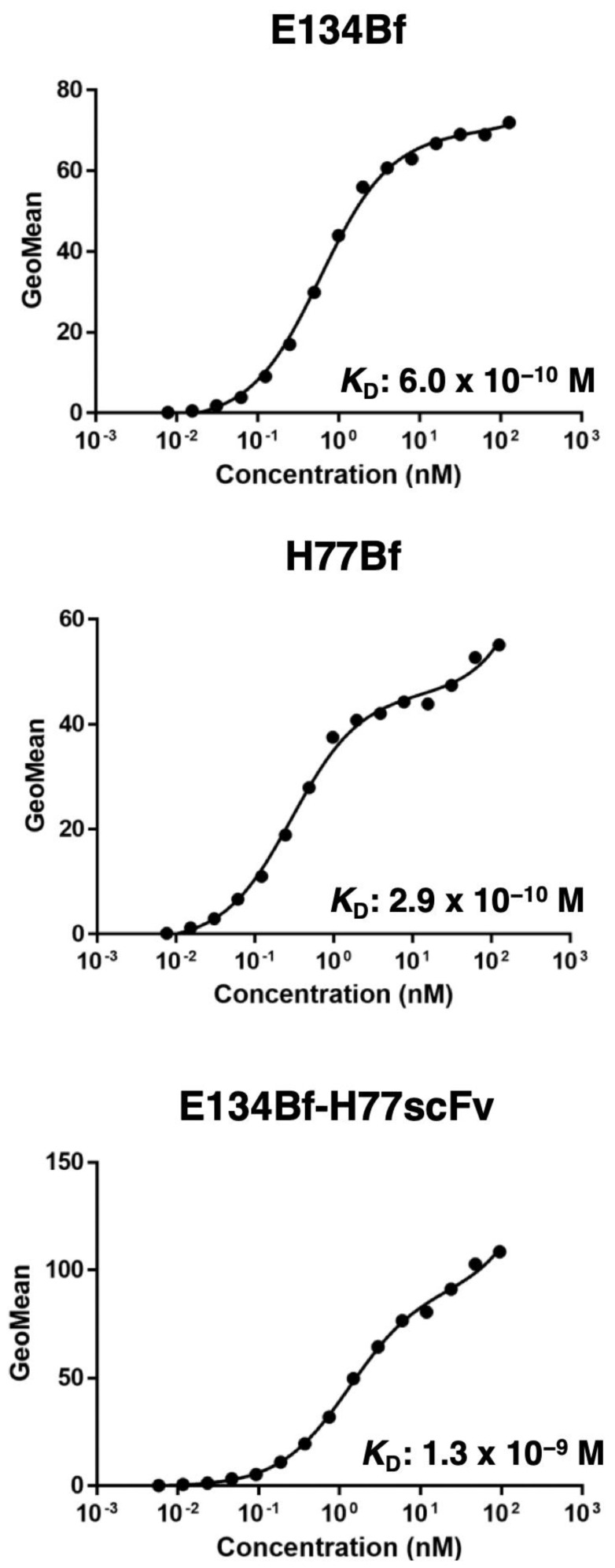
The determination of the apparent dissociation constant of E134Bf, H77Bf, and E134Bf-H77scFv for D-17 cells using flow cytometry. D-17 cells were incubated with serially diluted E134Bf, H77Bf, and E134Bf-H77scFv, followed by the treatment with Alexa Fluor 488-conjugated anti-dog IgG. Fluorescence data were analyzed using the Cell Analyzer EC800, following the calculation of the apparent dissociation constant (*K*_D_) by GraphPad Prism 8.

**Figure 4 pharmaceutics-14-02494-f004:**
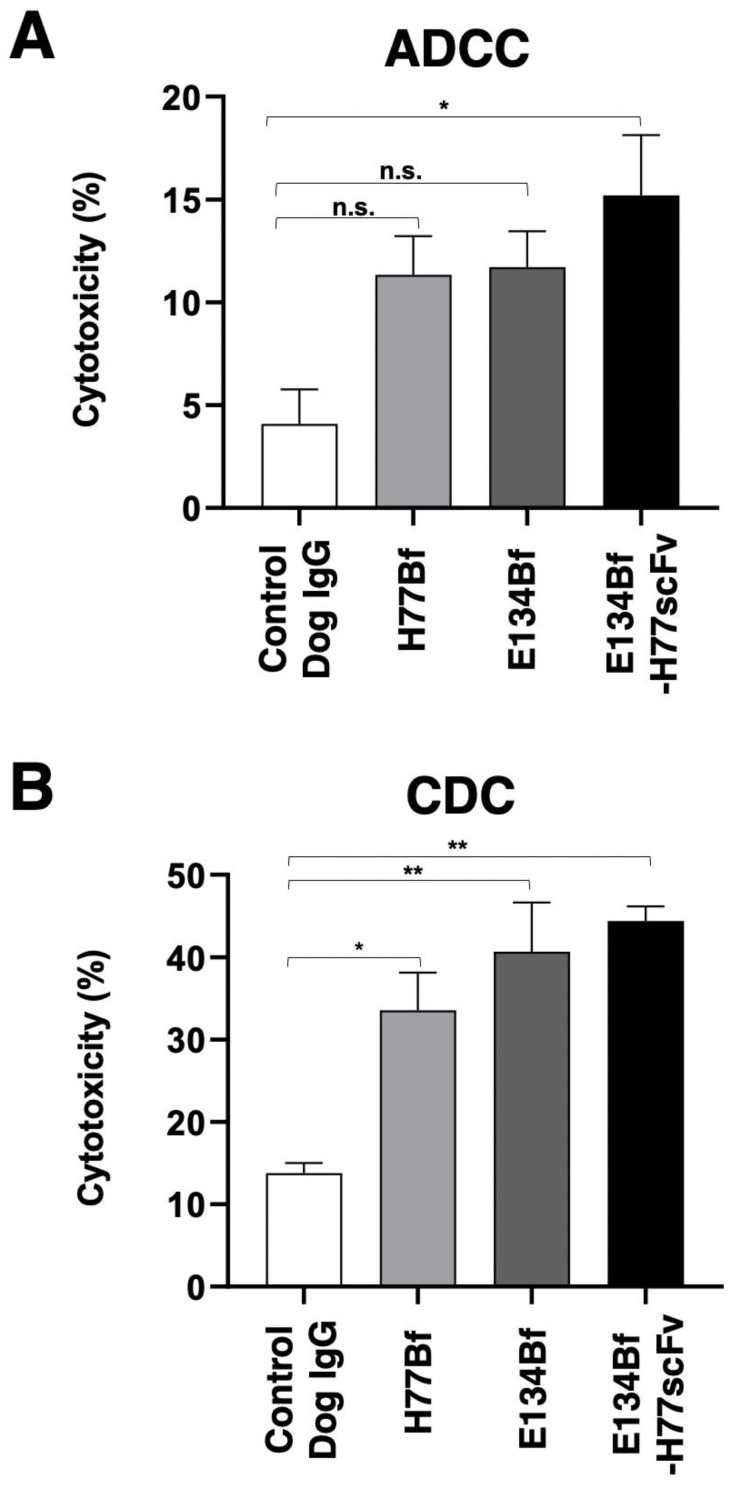
Evaluation of ADCC and CDC induced by E134Bf, H77Bf, and E134Bf-H77scFv. (**A**) ADCC by E134Bf, H77Bf, E134Bf-H77scFv, and control dog IgG against D-17 cells in the presence of canine MNCs. (**B**) CDC elicited by E134Bf, H77Bf, E134Bf-H77scFv, and control dog IgG targeting D-17 cells. Values are presented as the mean ± SEM. (** *p* < 0.01 and * *p* < 0.05; Welch’s *t* test). ADCC, antibody-dependent cellular cytotoxicity; CDC, complement-dependent cytotoxicity; MNCs, mononuclear cells; n.s., not significant.

**Figure 5 pharmaceutics-14-02494-f005:**
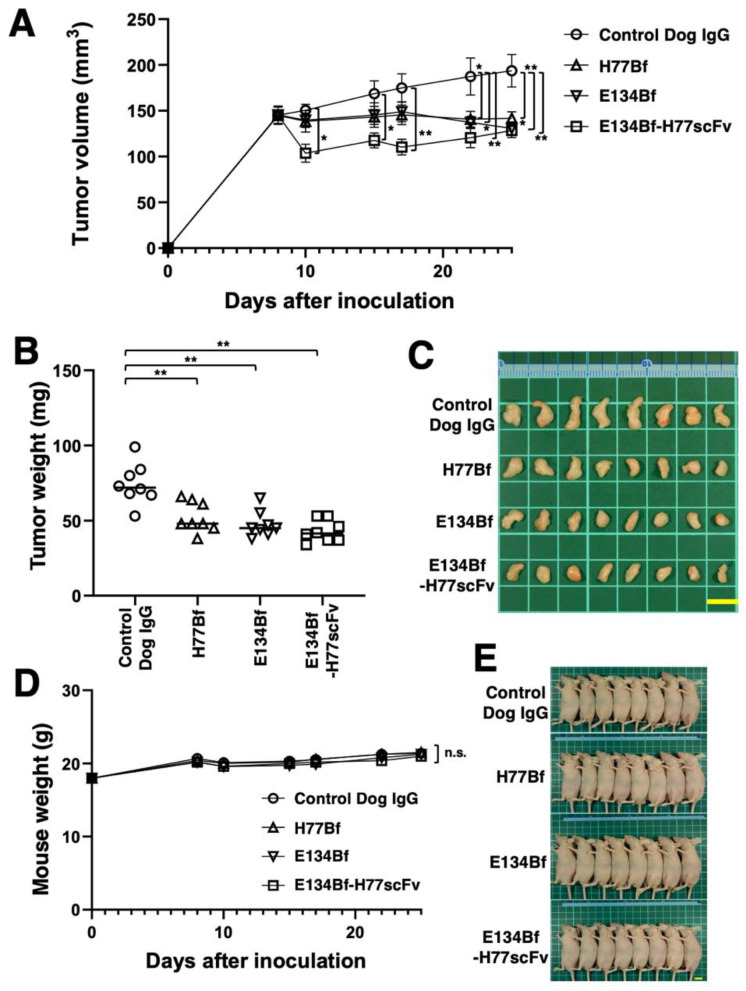
Antitumor activity of E134Bf, H77Bf, and E134Bf-H77scFv. (**A**) Evaluation of tumor volumes in D-17 xenograft. D-17 cells (5 × 10^6^ cells) were subcutaneously inoculated into mice. On day 8 after the inoculation of D-17 cells, 100 μg of E134Bf, H77Bf, E134Bf-H77scFv or control dog IgG were injected intraperitoneally. Additional antibodies were injected on days 15 and 22. Canine MNCs were also injected surrounding tumors on days 8, 15, and 22. The tumor volume was determined on days 8, 10, 15, 17, 22, and 25. Values are presented as the mean ± SEM. ** *p* < 0.01 and * *p* < 0.05 (ANOVA and Sidak’s multiple comparisons test). (**B**) Xenograft weight (day 25) was measured. Values are presented as the mean ± SEM. ** *p* < 0.01 (Welch’s *t* test). (**C**) Resected xenografts appearance of from the indicated groups on day 25 (scale bar, 1 cm). (**D**) Mice body weights inoculated with D-17 xenografts on days 8, 10, 15, 17, 22, and 25 (ANOVA and Sidak’s multiple comparisons test). n.s., not significant. (**E**) Body appearance of D-17- inoculated mice on day 25 (scale bar, 1 cm).

## Data Availability

The data presented in this study are available in the article and Appendix A.

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
