# Peer review of "Antitumor Activity of an Anti-EGFR/HER2 Bispecific Antibody in a Mouse Xenograft Model of Canine Osteosarcoma"

_pharmaceutics, 2022, doi:10.3390/pharmaceutics14112494_

Round 1

Reviewer 1 Report

Sarcomas are a large, heterogeneous family of rare (less than 2% of cancers in humans) and complex tumours (there are at least 50 types, with very different features and molecular characteristics). A distinction is made between soft tissue sarcomas on one hand and the rarer bone and cartilage sarcomas on the other hand. These cancers are very aggressive and the risk of recurrence is high. Pulmonary metastases are frequent and the prognosis is poor. Conventional treatments such as surgery, adjuvant radiotherapy and chemotherapy are not fully satisfactory. There is therefore an unmet medical need. In this context, the search for new therapeutic strategies is essential, including in the canine model, as stated by the authors (lines 73-75).

Here, the authors propose immunotherapy based on a high affinity (nM) bi-specific antibody directed against dog (d) EGFR and dHER2, overexpressed by stable transfected CHO cells and canine OS cell line D-17. The desired effector properties (ADCC and CDC) are well preserved, and the administration of this drug candidate suppresses the development of D-17 xenograft tumour in mice paving the way to new therapeutic strategies.

Overall, the manuscript is well written, well organized, and well-documented, fitting the scope of this special issue of Pharmaceutics. It complements two previously published studies, organized around a similar scientific approach, in which the properties of the two monospecific chimeric antibodies are reported (Cells 2021, 10:3599 and Oncol Rep. 2022, 48:xx). This third paper should be of interest to a general audience.

I would like to draw the authors' attention to the following points and suggest modifications that would clarify some points and help  the manuscript’s understanding

1- The antibody backbone of the chimeric bi-specific antibody should be indicated. Is it an IgG1, IgG2, IgG3 or IgG4? Ideally the sequence should be accessible. This point may be important for interpretation of ADCC and CDC experiments.

2- line 128: How is the fusion between H77 scFv and the E134B VL done? Is there any peptide linker between E134B constant light chain domain and the VH domain of H77scFv? This could influence the flexibility of the molecule with an impact on its antigen-binding activity. In addition, the choice of the fusion protocol should be discussed. Are there any advantages or disadvantages to fusing the scFv at the C-term of the light chain as compared to the N-term or C-term of the E134B heavy chain?

3- The chimeric molecule E134Bf-H77scFv is quite complex, and some degradation may occur upon expression in CHO cells. It would be valuable to get a direct quality analysis of the purified product (SDS-PAGE, WB or equivalent in the text or in supplementary material). Are there any co-purified degradation products in the drug substance that could compete in functional assays?

4- line 134: The Dog IgG is probably a polyclonal mixture of antibodies belonging to different IgG subclasses. This should be indicated. Indeed, as a control, a monoclonal antibody of the same subclass as the chimeric protein is preferred.

5- Figure 2: flow cytometry analysis: the experimental conditions, especially antibody concentrations, should be indicated.

6- Figure 3: What is measured in such conditions is not an affinity and must, at best, be qualified as an apparent affinity.

The discussion of this article could be revised and the following points addressed:

-       Chimeric protein design: Fusion of the scFv to the light chain. Does this provide any particular benefit?

-       The chimeric protein is tetravalent. However, there is no way to know how many of these valences are really working and engaged in antigen-binding. We do not even know whether the chimeric protein is able to cross-link dEGFR and dHER2. This may have a great impact on the biological effects.

-       It would be worth it to reiterate the limits of the D-17 xenograft model. Sarcomas are rare cancers (at least in humans) with a great diversity in terms of molecular characteristics so much so that it would be best to remain cautious with any generalizations of the approach proposed by this work, even if it is a well-conducted early pre-clinical approach.

Reviewer 2 Report

This paper described the cytotoxic activity of a IgG-scFV tetravalent construct on cancer cells expressing both dEGFR and dHER2 antigens: The results are interesting and well described. I believe that the paper would be strengthened by the addition of experiments as follows:

1) a competition experiment that could be done by a FACS analysis of the IgG-scFv in the presence of ECD or single epitope peptides to unravel the contribution of the single epitope to the increased activity of the constrtuct if compared with that of the relative antibodies

2) a Kaplan-Meyer curve should help in understanding the real curative potential of the bispecific tetravalent antibody format

Round 2

Reviewer 2 Report

The authors made a suggested measure. I can understand that the in vivo experiment is difficult to carry out, but this would have strengthened the paper.